# The Neurobiology of Love and Pair Bonding from Human and Animal Perspectives

**DOI:** 10.3390/biology12060844

**Published:** 2023-06-12

**Authors:** Sarah A. Blumenthal, Larry J. Young

**Affiliations:** 1Silvio O. Conte Center for Oxytocin and Social Cognition, Emory National Primate Research Center, Emory University, Atlanta, GA 30329, USA; 2Center for Translational Social Neuroscience, Emory University, Atlanta, GA 30329, USA; 3Department of Psychiatry and Behavioral Sciences, Emory University School of Medicine, Atlanta, GA 30322, USA

**Keywords:** pair bond, romantic love, prairie vole, oxytocin, dopamine, vasopressin

## Abstract

**Simple Summary:**

Being in love is a powerful emotional experience that is uniquely human; however, animal models of pair bonding provide insights into the neurobiological processes underlying love. Pair bonds are selective associations between two individuals (e.g., individuals in love) and can be studied in monogamous rodents such as prairie voles. Here, we examine pair bonds, from their evolutionary origins in mother–infant bonds, to the stages of bonding, comparing rodent and human literature. We discuss the neural circuits and neuromodulators driving bonding across species, with rodent studies providing insight into our human experiences of love.

**Abstract:**

Love is a powerful emotional experience that is rooted in ancient neurobiological processes shared with other species that pair bond. Considerable insights have been gained into the neural mechanisms driving the evolutionary antecedents of love by studies in animal models of pair bonding, particularly in monogamous species such as prairie voles (*Microtus ochrogaster*). Here, we provide an overview of the roles of oxytocin, dopamine, and vasopressin in regulating neural circuits responsible for generating bonds in animals and humans alike. We begin with the evolutionary origins of bonding in mother–infant relationships and then examine the neurobiological underpinnings of each stage of bonding. Oxytocin and dopamine interact to link the neural representation of partner stimuli with the social reward of courtship and mating to create a nurturing bond between individuals. Vasopressin facilitates mate-guarding behaviors, potentially related to the human experience of jealousy. We further discuss the psychological and physiological stress following partner separation and their adaptive function, as well as evidence of the positive health outcomes associated with being pair-bonded based on both animal and human studies.

## 1. Introduction

“How little we know, how much to discover, what chemical forces flow from lover to lover”—Frank Sinatra

Expressions of love have appeared in all forms of human creativity, from the arts to architecture, throughout history (e.g., Gustav Klimt’s The Kiss, William Shakespeare’s Romeo and Juliet, and the Taj Mahal). Love ensures successful reproduction, and our physiology relies on pair bonds to maintain mental and physical well-being [1,2]. Falling in love may be a uniquely human experience due to our advanced cognitive capabilities, but the underlying neurobiological processes involved in falling in love and maintaining long-term romantic relationships are most certainly rooted in ancient brain mechanisms predating the evolution of *Homo sapiens* [3]. Research using animal models of parental care, sexuality, pair bonding, and partner separation has provided considerable insights into the neural mechanisms driving evolutionary antecedents of love [4,5]. These findings allow for a greater understanding of our own human experiences of bonding and can inform strategies to engage in and maintain positive relationships. Romantic love involves several stages, beginning with attraction, followed by the formation of a bond, often facilitated by sexual intimacy, and finally, the maintenance of the bond over time [6,7]. These stages of relationship formation and maintenance are also seen in monogamous animals that form pair bonds [8,9,10].

Pair bonds refer to selective associations between two individuals of the same species [4]. These strong social relationships are typically observed within breeding pairs of monogamous species; however, pair bonds can exist between animals that are not sexually involved or sexually exclusive. Approximately 9% of mammals are socially monogamous, and pair bonding is often accompanied by shared territory, preferential mating, and affiliative displays between partners [4,11]. These elements of bonding typically outlast the course of a single mating cycle or season [12]. Although humans have remarkable flexibility in mating styles, from polygyny to serial monogamy, we have a strong tendency to pair bond, with social monogamy observed across diverse cultures throughout history [13]. However, biological studies of human bonding are limited in their ability to elucidate the underlying neural mechanisms culminating in bonding (Figure 1).

Much of our understanding of neural mechanisms responsible for pair bonding comes from studies in prairie voles (*Microtus ochrogaster*), a socially monogamous rodent. Discoveries made in prairie voles, along with findings from human imaging studies, have led to new insights into the neurobiology of romantic love [9,14]. Here, we review current rodent and human literature on bonding, focusing on oxytocin (OT), the mesolimbic dopamine (DA) reward system, and arginine vasopressin (AVP), as there is evidence of the involvement of these neurotransmitter systems in pair-bonding behavior in both animals and humans alike [15,16,17,18,19]. We begin with the evolutionary origins of bonding in mother–infant relationships and then discuss the progressions of a bond, from initial attraction, sex, and pair bond formation to long-term bond maintenance. There is still much to discover, but comparisons of animal models of bonding with the human experience of love continue to elucidate “what chemical forces flow from lover to lover”, rest assured Mr. Sinatra.

## 2. The Relationship between Parental Nurturing and Pair Bonding

### 2.1. Parental Bonding as the Evolutionary Antecedent to Pair Bonding

Holding your baby for the first time is a defining moment in a mother’s life. Mothers often experience an overwhelming sense of love and affection for their children and a strong desire to protect and provide for them. This moment is filled with relief, excitement, and joy in anticipation of the journey of parenthood to come. The mechanisms regulating mother–infant bonding are thought to be the evolutionary antecedent for mammalian pair bonding, and studies in rodents and humans alike point to the similarities between these types of bond [3] (Figure 2a). Infant mammals require considerable maternal care until at least the time of weaning, resulting in mothers investing time, energy, and resources into their offspring following birth. Mother–infant bonding enables the ongoing maternal drive to care for one’s young, thus improving long-term outcomes for offspring [5,20]. Maternal behavior in mammals emerges around the time of birth due to changes in circulating levels of the hormones estrogen, progesterone, prolactin, and OT [21,22]. Research in humans and in other mammals demonstrates a role for OT in promoting prosocial behavior, bonding, and the saliency of social cues [9,23,24]. OT is released into the bloodstream during birth in a pulsatile manner from hypothalamic neurons projecting to the posterior pituitary to stimulate uterine contractions and the ejection of milk during nursing [5]. Collaterals of those hypothalamic neurons release OT into the brain, resulting in neural circuit alterations contributing to the emergence of maternal behavior [25,26].

Mother–infant bonding is dependent on mothers recognizing infant stimuli through various sensory modalities and a motivational drive to care for their offspring [20]. For instance, pup retrieval in mice is enabled by increased attention to pup vocalizations as a consequence of OT-dependent alterations in the excitatory-inhibitory balance within the auditory cortex. OT thereby influences the signal-to-noise ratio of stimuli to increase the saliency of pup calls [27]. Rodent dams display promiscuous maternal behavior and will nurture any pup they encounter, likely because pups are altricial and restricted to the nest. In contrast, ungulates, such as sheep, give birth to mobile precocial offspring and thus must be able to discriminate the olfactory cues of their offspring from that of other lambs in the herd [5]. Ewes form selective bonds based on olfactory recognition of their offspring within 30 min of birth and will reject foreign lambs from nursing. Discrimination between olfactory cues from offspring versus alien lambs is facilitated by OT within the olfactory bulb, increasing the release of noradrenaline (NA), acetylcholine (ACh), and glutamate [28]. As in mother–infant bonding, selective recognition of an individual through sensory cues is essential for the bonding between mates, which will be discussed later. A persistent motivational drive to care for offspring is another essential component of the mother–infant bond. Following birth, activity in the mesolimbic dopamine (DA) system drives continuous interactions with offspring, including retrieval, licking, and grooming of pups [29,30]. This system comprises DA-producing neurons in the substantia nigra (SN) and the ventral tegmental area (VTA) that project to the striatum, cortex, limbic regions, and hypothalamus to influence motivation, emotion, and reward-related behaviors [31]. During mother-pup interactions, OT binds to oxytocin receptors (OXTRs) in the medial preoptic area (MPOA), which in turn leads to DA release from projections from the VTA to the nucleus accumbens (NAc), both regions implicated in motivation and reward [32,33].

Male involvement in parental care is relatively uncommon in mammals; however, paternal care is more prevalent in socially monogamous mammals [12]. Selection for monogamy likely evolved as a mating strategy for males in situations where breeding females are solitary, female density is low, and males are unable to defend access to multiple females across territorial ranges [34]. Increased paternal care is positively related to genetic monogamy, with 56% of the 226 recorded socially monogamous mammal species displaying paternal behavior, suggesting that male engagement in parental behavior is favored when there is increased certainty in the paternity of offspring [11]. Paternal behaviors are thought to have evolved from the neural substrates of maternal behavior through enhanced interactions with offspring, with females selecting for affiliative and nurturing traits in males. In early hominids, it is theorized that males adopted a reproductive strategy of habitually mating with a specific female throughout the menstrual cycle to ensure a greater probability of fertilization [35]. These males are also thought to have exchanged resources, such as food, with females for sex. Females who chose to mate with provisioning males decreased their time spent searching for food, subsequently allowing for increased infant care and avoidance of predators. The trend towards a monogamous mating style and the emergence of paternal involvement in early hominids led to decreased competition among males for access to females, along with increased cooperation and prosociality within mating dyads.

The emergence of paternal behavior, similar to maternal behavior, is driven in part by OT activity in the mesolimbic DA system [36,37]. In mandarin voles (*Microtus mandarinus*), paternal behavior is faciliated by the activation of OT projection neurons from the paraventricular nucleus of the hypothalamus (PVN) to the VTA and NAc [37]. In humans, intranasal application of OT (IN-OT) increases parenting behaviors that support parent-infant bonding in fathers toward their infants [38]. Fathers also exhibit increases in gray matter volume in regions involved in motivation, including the striatum, medial amygdala (MeA), and lateral prefrontal cortex (lPFC) [39]. Functional magnetic resonance imaging (fMRI) studies indicate that the mesolimbic DA system is activated as parents of both sexes view pictures of their own children but not pictures of other children, and this activation is enhanced by IN-OT [40,41,42]. In human mothers, behavioral synchrony between mother and infant is related to changes in endogenous DA and the functional connectivity of the NAc, MeA, and the medial (mPFC), together comprising the MeA network implicated in social functioning [43,44]. Additionally, plasma OT is correlated with the activation of the amygdala and NAc in mothers viewing their own infant [45]. Interestingly, individuals in long-term romantic relationships show similar activation of DA-rich rewards and social regions when viewing pictures of their partner [46,47]. In addition to the effects of OT and DA in promoting parental care of both sexes, AVP, known to promote territoriality, aggression, and mating, has been linked to parental nurturing [48,49,50,51]. In female rats, viral vector-mediated upregulation of the AVP 1a receptor (AVPR1a) within the MPOA improves maternal care, whereas blockade of the same receptor impairs maternal actions [49]. The influence of AVP on male parental behavior in prairie voles is localized to the lateral septum (LS), where local AVPR1a stimulation or blockade respectively increases or decreases paternal behavior [48]. Furthermore, plasma levels of AVP in human fathers, but not mothers, are related to activation of the inferior frontal gyrus and in the insular cortex (IC), both regions implicated in socio-cognitive function, suggesting a distinct role for AVP in promoting paternal behavior [45].

Maternal bonding establishes a system whereby the mother is attuned to cues from their offspring and has a persistent motivation to care for their young. In males, similar processes, such as attraction to infant-related stimuli, emerge in animals adopting a monogamous mating and social strategy [11,52]. Similar neural circuits responsible for parent-infant bonds drive the development and maintenance of bonds between mating pairs of socially monogamous individuals. From parental behavior evolved a sensitivity to partner cues (i.e., heightened salience mediated by OT) and a persistent motivation to be around one’s partner (i.e., DA activation of reward/motivation circuits). Together, these processes aid in maintaining pair bonds over long periods of time. Furthermore, parent-offspring interactions in monogamous mammals set the stage for later bonding in adulthood.

### 2.2. The Influence of Parental Care of Offspring in Later-Life Bonding

Attachment theory, first put forth by Mary Ainsworth and John Bowlby, suggests that early social interactions with caregivers shape representations of self-versus other and affect relationship formation and maintenance into adulthood [53,54,55]. Early in life, individuals seek out caregivers to aid during times of need, distress, or danger, resulting in socially-dependent regulation of homeostasis [56,57,58]. When proximity seeking regularly leads to attention from the caregiver and proper homeostatic maintenance, secure attachment emerges. Alternatively, when caregivers are absent, or proximity seeking does not alleviate distress, some develop insecure attachment strategies. Insecurely attached individuals learn to self-soothe instead of relying on co-regulation with caregivers and distance themselves from attachments with others [59]. Parent–child attachments may influence aspects of later romantic relationships, as attachment security in early life has been associated with greater romantic relationship satisfaction in adulthood [60]. Studies using laboratory-based conflict discussions between romantic partners show that increased quality of parent–child interactions during early life predict positive interactions towards one’s romantic partner (e.g., better conflict resolution, less hostility, more warmth), whereas low-quality parent-child relationships predict negative interactions (e.g., hostility, anger, manipulation) [60]. However, a variety of additional factors contribute to adult romantic attachment styles, including interpersonal peer relationships, romantic history, and genetics [61]. Studies in prairie voles illustrate how early parental care can influence later social behavior and bonding in adulthood. In the past few decades, prairie voles have emerged as a model species for understanding monogamous mating strategies and complex social relationships as they form enduring pair bonds with an opposite-sex partner and exhibit biparental care for their offspring [4,9,62]. Voles receiving high parental care spend more time nursing and huddling with parents than those receiving low parental care [63]. In adolescence, high-care animals investigate novel conspecifics more, whereas those receiving low early-life parental care display increased autogrooming, a self-soothing behavior, in the presence of novel conspecifics. Voles raised by their mother alone, compared to those raised under biparental care, receive less parental nurturing in the form of licking/grooming early in development [64]. In adulthood, these same animals are slower to form a preference for their partner, despite normal mating behavior. Furthermore, voles raised by a single mother display decreased turnover of DA in the NAc following cohabitation and mating with an opposite-sex conspecific, suggesting less activity in reward regions during sexual contact [64,65]. In one study, prairie vole pups were isolated from their parents and littermates for 3 h per day for the first two weeks of life, then raised normally by both parents [66]. As adults, females that experienced this early-life separation, modeling parental neglect, displayed an impairment in the ability to form pair bonds. Interestingly, not all females responded in the same way to this early neglect; some were resilient, forming bonds normally, while others were susceptible and unable to form bonds. Further investigation revealed that those females with the highest density of OXTR in the NAc were resilient to early-life neglect and formed bonds, while those with lower OXTR density were impaired in pair bond formation. It is hypothesized that the extra nurturing received by the pups upon reuniting with the parents after the separation resulted in OT release in both high and low-OXTR density animals, but the high OXTR animals experienced more receptor signaling, effectively protecting against the effects of the 3 h social isolation.

Early-life nurturing is associated with epigenetic modifications to the oxytocin receptor gene (*Oxtr*) that may explain later differences in social behavior in adulthood. The MT2 region of *Oxtr* is highly conserved across prairie voles and humans, and DNA hypomethylation at this site is associated with increased OXTR expression [67]. Prairie voles receiving lower levels of parental care have hypermethylation of the MT2 region, which is predictive of lower OXTR expression in the NAc [67,68]. Epigenetic alterations to *Oxtr* may, in turn, affect social behavior later in life. Similar results are seen in humans, as children experiencing childhood abuse or neglect have hypermethylation of the *Oxtr* MT2 region in the orbital frontal cortex (OFC) [69]. Additionally, genetic polymorphisms in the vole *Oxtr* robustly predict NAc OXTR density, which predicts resilience to parental neglect. The C/C, C/T, and T/T variants of the single nucleotide polymorphism NT213739 of *Oxtr* predict high, medium, and low OXTR expression in the NAc, respectively [70]. These genetic variations may also influence social behavior, as male C/C prairie voles display enhanced social attachment as compared to C/T and T/T animals [71]. Furthermore, early life experience may interact with these genetic polymorphisms and affect later life social behavior. As mentioned previously, prairie voles raised with biparental care, compared to those raised by a mother alone, form more pronounced pair bonds in adulthood. C/T animals display more robust preferences for their partner overall, compared to T/T voles, whereas T/T animals show more pronounced effects of rearing on later partner preference [70]. Together, these findings from studies in prairie voles provide evidence that early-life nurturing influences later life-bond formation through epigenetic mechanisms and genetic variation in *Oxtr*. Continued investigations into the effects of early-life care on social behavior in adulthood, through animal models of bonding and human studies of attachment and relationship quality, are necessary to understand the neural and psychological mechanisms beginning early in development that prime later social relationships.

## 3. Stages of Bonding

How do individuals go from strangers to lovers? Although no two relationships are alike, most undergo similar progressions. When people are still strangers, a relationship begins with attraction followed, in time, by a decision to pursue a relationship with an individual [72,73]. As the relationship begins to deepen and familiarity grows a bond forms, and people may experience feelings of being in love [74]. The formation of this bond is facilitated by the emotional and physical connection that sex provides, which due to individual differences and preferences, may initially occur at different time points in the relationship progression. Loving romantic relationships can last throughout a lifetime, requiring continued commitment and communication to maintain, and when these relationships end, they can pose threats to psychological and physiological well-being [75]. Although the loss of a relationship can be devastating, there is evidence of long-term health benefits of being in a committed partnership [76,77]. Research on the underlying neurobiological mechanisms of mating behavior and pair bonding in prairie voles, as well as in other rodents, may provide useful translational insights into human experiences of love (Figure 2). Importantly, a considerable amount of overlap exists in the brain regions and neuromodulators involved in these processes in rodents and humans. Many of these regions are a part of established networks known to be involved in social behavior across species, such as the social brain network (SBN) and the social decision-making network (SDMN). The SBN, described in rodents, birds, and reptiles, comprises hypothalamic structures as well as the periaqueductal gray (PAG), MeA, bed nucleus of the stria terminalis (BNST), and the LS that are reciprocally connected with the other regions of the network, contain receptors for gonadal hormones, and are implicated in at least one social behavior [78,79]. This network was later expanded as the SDMN to include structures in the mesolimbic DA system involved in detecting and appraising the rewarding aspects of social situations [80]. Many of the structures of the SBN and SDMN have been implicated in human social function [43,46,81]. Cortical regions involved in emotional processing and decision-making, including the mPFC, ACC, and IC, have also been implicated in social function in rodents and humans alike [81,82,83]. The involvement of these conversed networks of social behavior across species in attraction, mating, and bonding further illustrates the translational applicability of rodent research investigating these processes, which will be discussed further in this section.

### 3.1. Sexual Attraction and Mate Choice

A glance across a table, a look from the other side of the room, or a picture on a dating application; love may not always happen at first sight, but attraction often does. Individuals make quick subjective decisions on whether to approach and pursue an attractive person, and these decisions are accompanied by feelings of exhilaration and craving for a potential romantic connection. Our initial evaluations of another’s attractiveness are often primarily influenced by appearance, although many factors influence the decision of whether to pursue a relationship with another individual, such as personality, compatibility, and shared values [84]. Since evolutionary fitness is determined by the successful reproduction and survival of offspring, species-specific mechanisms for attracting and selecting appropriate mates have evolved.

In animals, mate choice is driven by the initial attraction of potential mates and the evaluation of their reproductive fitness through various modalities [72,85]. For rodents specifically, olfactory and auditory cues from an opposite-sex conspecific drive approach behavior and are used to attract potential mates (Figure 2b). Olfactory cues carry information about sex, strain, social rank, health, and sexual receptivity [86,87]. Olfactory cues are also used to remember individuals, a process essential for forming a pair bond. OT plays an important role in processing olfactory signals involved in assessing partner quality and social recognition [87]. For example, female mice display a preference for odors from healthy males over those with an infection; however, this preference is diminished in female mice lacking OXTRs. During social interactions, OT modulates sensitivity to olfactory cues through activity at OXTRs in the anterior olfactory nucleus (AON). OT in the AON facilitates an increased inhibitory tone in the main olfactory bulb (MOB), resulting in a greater signal-to-noise ratio for odors associated with social interactions and subsequently leading to a higher quality signal being transmitted to downstream regions [9,88]. Social recognition of a potential partner is then encoded as the neural representation of partner cues in the hippocampus [89]. Alongside olfactory cues, ultrasonic vocalizations (USVs) provide information about sexual history and fitness and are used to attract potential mates [90]. Male mice emit USVs in a sing-song pattern upon encountering a female, and females approach these vocalizations. Additionally, upon hearing male USVs, female mice show activation of kisspeptin neurons in the arcuate nucleus responsible for driving reproduction, providing a mechanism whereby male auditory cues promote fertility and approach behavior in females [91]. Similarly, male mice alter behavior in response to female USVs, and these calls contain information about the female’s reproductive receptivity [92,93]. In addition to olfactory and auditory cues, somatosensory cues detected through social tactile stimulation play an important role in interactions between animals. Social touch, as opposed to non-social touch, results in stronger responses from cells within the rodent barrel cortex, a division of the primary somatosensory cortex corresponding to processing signals emerging from whiskers [94]. Social touch also leads to the release of OT, from the PVN, and DA, from the VTA, to the NAc, resulting in conditioned place preference for social touch contexts and suggesting reinforcing properties of social tactile stimulation [95,96]. Importantly, olfactory, auditory, and somatosensory cues promote courtship in multi-modal and non-redundant ways, with the combination of the modalities resulting in greater courtship displays for both males and females as compared to either modality alone [90,97].

In humans and non-human primates (NHPs) alike, visual information about potential mates is favored over other sensory modalities, although there is evidence of auditory and olfactory information also shaping attraction [90] (Figure 2b). In primates, OXTRs are concentrated in visual attention and processing regions, such as the nucleus basalis of Meynert, superior colliculus, and primary visual cortex. Similarly, in humans, visual processing of social cues is facilitated by IN-OT through increased attention to and recognition of emotional cues in faces [98,99]. IN-OT also increases the activity of regions involved in visual processing and reward, along with the amygdala, a region responsible for assigning valence to social interactions [100]. Although perceived attractiveness is subjective and varies by culture, certain features are generally favored and may point to evolutionarily driven preferences for features indicative of reproductive fitness. For example, features favored include youthfulness, symmetry, and averageness (how closely a face resembles the majority of other faces), which are hypothesized to be related to the genetic diversity and health of the individual [101]. Facial attractiveness has also been found to be predictive of longevity and reproductive success [102,103].

fMRI studies assessing region-specific activation during initial attraction provide insight into how humans process the face of an attractive individual they have not previously met. Participants shown images of novel attractive opposite-sex individuals had rapid activation of the NAc, suggesting rewarding properties of viewing attractive faces [104]. Facial attractiveness is additionally evaluated by a network of cortical regions, including the lateral OFC, ventromedial PFC (vmPFC), and the anterior IC (aIC), with the degree of activity in these regions correlating with perceived facial attractiveness, which is then translated into a romantic interest [104,105]. In an fMRI task where participants were shown photos of opposite-sex individuals, activity in the dorsomedial PFC (dmPFC) predicted whether the participant was romantically interested in the person from the photo upon meeting them at a later real-life “speed dating” event [73]. Following initial attraction, learning that feelings of attraction or romantic interest are reciprocated promotes continued engagement. Upon learning of one’s own desirability, the posterior temporal sulcus, mPFC, aIC, and the dorsal anterior cingulate cortex (dACC) are activated; however, following rejection from a potential romantic interest, the aIC and dACC are also recruited [106,107,108]. These overlapping regions are involved in salience processing and evaluating the emotions of others, thereby contributing to updating romantic interest following acceptance or rejection [109,110]. Although visual information is utilized over other sensory modalities for initial attraction, auditory and tactile cues carry information about potential mates. Humans can typically decode the gender of an individual from their voice and distinguish between voices [111,112]. Social touch, on the other hand, conveys information about intimacy, warmth, and sexuality, with the frequency and bodily location of social touch indicating the closeness between individuals [113]. Both auditory and tactile social cues rely on the involvement of the mPFC and IC, implicated in socio-emotional and socio-salience processing [114,115]. A recent study involving real-life dating interactions between participants interestingly found that attraction was not associated with overt social signals between individuals, such as eye gaze or laughter, but rather with synchrony in heart rate and skin conductance between individuals [116]. This suggests that synchronous arousal contributes to feelings of attraction toward another person in a non-laboratory setting where decisions on attraction are not limited to single sensory modalities (i.e., picture or voice recording of an individual). Initial attraction is, therefore, driven by different sensory modalities in rodents and humans; however, the information gathered from these modalities helps inform future choices about pursuing a particular individual for reproduction or for a romantic relationship. Importantly, attraction and mate choice in humans is subjective and often further influenced by socio-cultural expectations for relationships, early life experiences, and previous relationship experiences [60,117,118].

### 3.2. Mating Activates OT, DA, and AVP Circuitry

Attraction and mate choice often culminate in sex, which for humans is an intimate social experience comprised of both physical and emotional connections. During sex, there is a sharing of and responding to social cues that can facilitate bonding between individuals (Figure 2c). People engage in sex for a variety of reasons beyond just procreation, including for pleasure. The appetitive and consummatory aspects of sex are regulated by gonadal steroids (estrogen, progesterone, testosterone) as well as by OT, DA, and AVP [51,119,120,121,122]. Sex, in turn, activates these systems to promote subsequent behaviors, including pair bonding. In rats, OT release from neurons in the PVN facilitates male sexual activity and ejaculation as well as mating-induced anxiolysis [122,123]. In prairie voles, OT signaling stimulated by mating enhances the coordinated activity of multiple brain regions in a social salience network, a grouping of interconnected nodes thought to encode the valence and incentive salience of socio-sensory cues. OXTR signaling in the NAc acts as a hub for coordinating activity among AON, PVN, BLA, and PFC [124]. DA also facilitates sexual behavior in both males and females. Stimulating DA receptors in the MPOA of male rats, a region critical for male sexual behavior, facilitates copulation, whereas blockade of these same receptors impairs sexual behavior [119]. In female rats, extracellular concentrations of DA in the NAc increase following exposure to a sexually active male and during copulation [125]. Rodent studies of mating suggest a sexually dimorphic role of AVP in sexual behavior. In male but not female mice, AVP cells in the BNST and MeA are active during mating, and the knockdown of BNST AVP cells inhibits male sexual behaviors [48,51].

Mating also increases the functional connectivity between the mPFC and NAc of female prairie voles [126]. The degree of local field potential (LFP) coupling between the mPFC and NAc after the first mating encounter predicts latency to initiate affiliative huddling with a partner, a behavioral characteristic of pair-bonded voles. Entrainment of mPFC and NAc LFPs may engage plasticity mechanisms in the NAc and increase neural responses to partner-specific neural engrams from nodes of the social processing networks during mating, including in the BLA and hippocampus [9] (Figure 3a). Additionally, dendritic spine density, a measure of neuroplasticity, is altered in both the mPFC and hippocampus of male rats following sexual activity, suggesting a potential mechanism for strengthening the association between memories of partner cues and reward [127,128].

Assessing brain regions involved in mating in human subjects is constrained by technical, ethical, and logistical considerations; however, sexual arousal has been extensively examined using fMRI. A large degree of overlap exists between regions highlighted in rodent studies of mating and human studies of arousal. Both men and women experiencing visually-induced sexual arousal show increased activity in the amygdala, hippocampus, mPFC, NAc, OFC, ACC, and IC [129,130,131,132]. During sexual self-stimulation, for both men and women, activity in the cerebellum, ACC, VTA, and NAc steadily increases until orgasm is reached [133,134]. In women, activity is also reported in the PFC, IC, hippocampus, amygdala, and hypothalamus [135]. OT plasma levels have also been reported to increase in both men and women during sexual arousal, peaking during orgasm [136]. Overlap in regions involved in sexual activity and rodents and sexual arousal in humans suggests conserved neurobiological mechanisms driving mating behavior.

### 3.3. Pair Bonding

In both humans and rodents alike, mating facilitates but is not necessary for the formation of a pair bond [4]. For humans, societal and cultural expectations, along with personal preferences, to delay sexual intercourse until marriage results in many successful pair-bonded relationships in the absence of sexual activity. Humans experience pair bonds as committed romantic relationships accompanied by feelings of attachment, euphoria, trust, and comfort, which all culminate in the experience of being “in love” (Figure 2d). In prairie voles, mating and cohabitation with an opposite-sex partner increases the preference for that partner as assessed by the partner preference test, which is used to measure the strength of a pair bond [137] (Figure 1a). In this test, an animal is placed into a three-chamber enclosure where they are able to roam freely. Their partner is tethered in one chamber, a stranger (same sex as the partner) is tethered in another chamber, and the middle chamber is left unoccupied. The time spent in each chamber and time spent huddling with the partner or stranger is quantified. Pair-bonded prairie voles spend more time with their partner than with the stranger, whereas non-monogamous meadow voles (*Microtus pennsylvanicus*) spend little time interacting with their partner or stranger [138]. Pair bonding is regulated by OT, DA, and AVP, and it has been postulated that pair bonds result from synaptic plasticity mechanisms initiated during positive social interactions, such as mating [9] (Figure 3a). These interactions may therefore strengthen the association between neurons encoding the identity of the partner and circuits regulating reward and reinforcement.

Oxytocin has been studied for its involvement in the prairie vole pair bond, in part due to the high density of OXTRs in the prairie vole NAc and mPFC, as compared to other non-monogamous vole species [139]. In female prairie voles, pharmacological blockade of OXTRs in either the NAc or mPFC prevents mating-induced partner preference [9]. Additionally, viral knockdown of OXTRs within the NAc of voles disrupts partner preference formation and alloparental behavior; however, pair bonding and maternal care are facilitated when OXTR expression in the NAc is increased, suggesting a direct role of NAc OXTRs in driving affiliative behavior [15,140]. Mating strengthens functional connectivity between the mPFC and NAc and predicts affiliative behavior towards the partner [126]. Furthermore, rhythmically activating this circuit during social interactions drives pair bond formation, and this may be facilitated by OT activity within this circuit.

Social interactions in rodents stimulate the release of OT from the PVN in the NAc and VTA, and OT binding in the VTA leads to the release of DA in the NAc, thus contributing to the rewarding properties of social stimuli [16,26] (Figure 3a). Mating increases DA transmission in the NAc, and blocking DA receptors in the NAc disrupts the formation of a partner preference in voles, whereas activation of DA receptors facilitates pair bonding even in the absence of mating [141,142]. Interestingly, the effects of DA 1-type receptors (D1R) and DA 2-type receptors (DA2R) on bonding are opposing, with the activation of D1Rs inhibiting pair bond formation and the activation of D2Rs stimulating pair bonding [9,141]. Early studies in prairie voles determined that pair bonds typically emerge after 24 h of cohabitation with mating; however, stimulation of D2Rs in the rostral shell of the NAc in prairie voles was found to elicit a partner preference after only 6 h of cohabitation and in the absence mating [141,143]. The role of D1R receptors in pair bonds will be discussed in the next section on pair bond maintenance.

AVP also mediates mating-induced pair bonds. Male prairie voles administered an AVPR1a antagonist prior to cohabitation, and mating with a female vole failed to form a pair bond; however, administration of the same antagonist immediately following cohabitation with mating had no effect on later bonding [144]. AVP, like OT, displays marked differences in its receptor densities between monogamous and non-monogamous voles in regions involved in social behavior. Prairie voles show much higher AVPR1a expression in the ventral pallidum (VP) than meadow voles and viral knockdown of AVPR1a in the VP of prairie voles impairs partner preference [17,145]. Conversely, viral-mediated increases in VP AVPR1a expression induce partner preferences and increase affiliative behaviors in male prairie voles that are sexually naïve and, interestingly, also lead to partner preference formation in non-monogamous meadow voles [138,146]. These manipulations demonstrate that AVPR1a expression within the VP facilitates pair bonding. Furthermore, investigations into OXTR and AVPR1a expression across monogamous and non-monogamous rodents provides cross-species comparisons for neural underpinnings of pair bonding [139,147,148].

Genetic polymorphisms in the human *OXTR* and *AVP1RA* genes have been linked to bonding and relationship outcomes. Women with the GG genotype of the rs53576 *OXTR* polymorphism display enhanced empathy, sociability, and emotional stability while also reporting greater marital satisfaction than those with the AA or AG genotype, suggesting genetic variations in *OXTR* expression may influence human social behavior [149]. The RS3 repetitive polymorphism of the *AVPR1A* gene is associated with bonding in men, with those carrying one or two copies of the 334 alleles scoring lower on the Partner Bonding Scale than men who do not carry the 334 alleles [19]. The 334 alleles are also associated with an increased likelihood of having at least one instance of a marital crisis, decreased marital quality as assessed by one’s spouse, and decreased likelihood of being married. Continued investigations into the association between genetic polymorphisms and social relationships allow researchers to bridge the gap between rodent and human studies on the mechanisms underlying bonding.

fMRI studies on bonding and love in humans provide further insights into the regions driving these social experiences and consistently implicate DA-rich reward regions, similar to findings in prairie voles (Figure 3b). When individuals view images of their romantic partner or have romantic thoughts about their partner, the NAc and VTA are recruited [46,74]. In one study, IN-OT caused men in monogamous relationships to rate their partner as more attractive than when given a placebo, but IN-OT did not increase the perceived attractiveness of other women. fMRI revealed that IN-OT in combination with viewing the partner enhanced activation of the NAc and VTA consistent with the hypothesis of pair bonding based on vole studies [18] (Figure 3b). Furthermore, spatial activation patterns in the NAc can be discriminated between viewing one’s partner and viewing an opposite-sex close friend, suggesting that neural representations specific to the partner may exist, akin to the putative neural engrams encoding partner cues in the NAc, BLA, and hippocampus of rodents [150]. Other regions involved in emotional processing are activated when viewing or thinking about a long-term partner. These include the IC and ACC, which are both associated with complex social processes such as empathy and in integrating social, sensory, and emotional information [46,47]. The caudate and putamen also show activation upon exposure to partner stimuli, and these regions are involved in emotional processing, motivation, and reward, as well as the execution of movement. Viewing pictures of one’s partner leads to the deactivation of several areas as well, including the amygdala and PFC. Furthermore, several regions involved in romantic love overlap with those involved in maternal love, including activation of the IC, striatum, and SN, and deactivation of the right mPFC and amygdala, among other regions involved in attachment and reward, suggesting conserved circuitry of bonding across different types of relationships [46].

### 3.4. Pair Bond Maintenance

Bonds between socially monogamous animals are maintained over long periods of time, usually lasting more than one mating cycle and, in many cases, sustained throughout the lifetime [4]. Maintaining a pair bond involves preventing one’s partner from bonding with another individual and negative reinforcement from being away from the partner. Together these processes prevent other potential mates from disrupting an existing pair bond, and reward continued proximity to the partner.

#### 3.4.1. Preventing Disruption of an Existing Pair Bond

The physical and emotional intimacy of being in a monogamous romantic partnership is supposed to be shared only between those within the relationship. This is what sets romantic partnerships apart from relationships with friends and family and why both physical and emotional cheating can be devastating. In a healthy relationship, trust that your partner is committed is essential; however, humans also exhibit behaviors to prevent individuals from disrupting an existing romantic partnership. So, how can you ensure that your partner only has eyes for you? We may experience feelings of jealousy when seeing our partner with an attractive person, perhaps culminating in aggressive behaviors. Alternatively, humans in committed relationships may try to avoid temptation from attractive people by devaluing the attractiveness of potential mates. These behaviors are useful for maintaining a bond long-term when alternative options become available. Interestingly, pair-bonded prairie voles similarly exhibit aggression towards extra-pair individuals to aid in pair bond maintenance.

Although prairie voles are socially monogamous, they are not exclusively sexually monogamous, with some males and females seeking out extra-pair copulations [151,152]. Male prairie voles in naturalistic settings can be typically divided into three categories, exclusively mating bonded males, non-exclusive mating bonded males, and unpaired wanderer males [152]. Exclusively mating bonded males may shift to a non-exclusive mating style in certain social and environmental contexts to increase offspring production; however, by leaving their female mate, they risk cuckoldry by other nearby males [153]. To prevent their partner from mating with another male and to prevent another female from threatening an existing bond, pair-bonded males will develop selective aggression towards strangers following pair bond formation (Figure 2e). Sexually naïve male prairie voles exhibit relatively low levels of aggression directed at strangers, but following two weeks of cohabitation, mate guarding emerges. AVP has been associated with territorial aggression in other species, and activity at AVPR1as in the anterior hypothalamus (AH) specifically promotes mate guarding in prairie voles [154]. AVPR1a density in the AH is increased in pair-bonded male voles over sexually naïve males, and viral-mediated overexpression of AVPR1a in this region increases mate guarding. Furthermore, in female rats, mate guarding is accompanied by increased activity in AVP-producing neurons in the PVN [155]. Selective aggression toward unfamiliar females from male prairie voles is also believed to be facilitated by changes in D1R density in the NAc following pair bond formation. After two weeks of cohabitation with a female, male voles have a 60% increase in NAc D1R expression, whereas D2R density remains unchanged. DA transmission in the NAc shell is also elevated in pair-bonded voles, suggesting a continued role for DA in pair-bond maintenance [141,142]. Furthermore, blocking D1Rs in the NAc in pair-bonded males disrupts mate guarding. D1R activity in the NAc may act to prevent bonding generally, as males given a D1R agonist to the NAc failed to form a partner preference [141]. Thus, in pair-bonded voles engaging in occasional extra pair copulations, DA releasing during sex stimulates D1R to a greater extent than in single males, which may prevent new pair bonds from forming. In addition to changes in D1R density, the NAc sees an upregulation of k-opioid receptors (KORs) following pair bonding in male and female voles. KORs in the shell of the NAc encode social aversion to novel conspecifics, and blockade of KORs in the NAc shell diminishes selective aggression from both males and females [156]. Opioid signaling at KORs occurs downstream of NAc shell DA activation of D1Rs to generate selective aggression, whereas affiliative behavior remains unchanged following manipulations to NAc D1Rs and KORs [156,157]. Together, AVP, DA, and opioid signaling promote aggressive behaviors directed toward strangers to prevent other individuals from threatening an existing bond.

NHP models of monogamy can also be useful to elucidate regions involved in selective aggression. Coppery titi monkeys (*Callicebus cupreus*) pair bond with opposite-sex mates and display partner preference and mate guarding. Following exposure to their partner interacting with opposite-sex conspecific, male titi monkeys display increased activation in the LS and cingulate cortex along with elevated plasma testosterone and cortisol levels [158]. Furthermore, male and female titi monkeys display coordinated defensive and mate-guarding behaviors upon exposure to a simulated intruder [159]. Selective aggression serves to maintain pair bonds in human relationships and may best be compared to feelings of jealousy. There are few studies examining brain regions involved in jealousy; however, fMRI reveals that individuals with higher scores on a jealousy scale have increased activity in the IC and ACC and enhanced functional connectivity between the inferior frontal gyrus and dorsal striatum in response to threatening faces [160,161]. Humans also maintain bonds by devaluing the attractiveness of potential partners once in a romantic relationship; however, this relies on proper executive control and is diminished in the presence of enhanced cognitive load (Figure 2e). In one study, single and monogamous individuals were shown images of their partners and of attractive individuals. The right vlPFC and the posterior dmPFC increased activation during the devaluation of attractive individuals, with right vlPFC activation correlated with participants’ level of investment in their relationship [162,163]. The vlPFC is involved in emotion regulation and executive function, and its recruitment in devaluation is seen in long-term, but not new, romantic relationships [163]. This suggests that executive control over potential partner devaluation is a function of long-term relationship maintenance that is not required in the passionate early stages of romantic relationships. In another study, IN-OT or placebo was given to single men and men in long-term relationships, and the proximity between the male subjects and an attractive female confederate was examined [164]. Men in relationships kept a greater distance from the female confederate than single men, and this was enhanced by IN-OT. Interestingly, male subjects in long-term relationships were also slower to approach photos of attractive women, as compared to single male subjects, and IN-OT increased this effect. OT may therefore aid in pair bond maintenance by not only increasing the rewarding aspects of one’s partner but also by promoting avoidance of attractive alternative mates.

#### 3.4.2. Negative Reinforcement Drives Maintenance

The rewarding effects of partner cues help drive pair bond formation; however, the negative effects of partner separation facilitate the maintenance of long-term bonds. Love letters, longing, and a dramatic reunion at the airport (if you are in an early 2000s romantic comedy) are the result of an intense drive to be back with one’s partner after spending time apart. This negative reinforcement may have evolved to drive individuals to seek out their partner after being separated rather than seeking out another mate. Much of our understanding of the mechanisms behind partner separation comes from prairie voles, elucidating how the OT system interacts with innate stress responses following separation from a partner. Upon separation from a pair-bonded partner, male and female prairie voles display depressive- and anxiety-like phenotypes, increased sensitivity to pain, and physiological changes, including elevated heart rate and increased circulating levels of corticosterone and adrenocorticotropic hormone [165,166,167] (Figure 2f). The neuropeptide corticotropin-releasing factor (CRF) mediates behavioral and endocrine responses to stress and acts on CRF type 1 (CRF1) and CRF type 2 (CRF2) receptors [168]. Administration of a non-selective CRF receptor antagonist blocks depressive- and anxiety-like behaviors in male prairie voles separated from their partner, whereas infusions of a CRF2 agonist induce depressive-like behaviors in non-separated males, thus implicating CRF in mediating passive stress-coping following partner loss [166]. OT also plays a role in responses to partner loss, as infusions of OT to the NAc reduce passive stress-coping of separated males. Interestingly, CRF mRNA is elevated in the BNST of pair-bonded male voles, as compared to non-paired males, and CRF2s colocalize to OT neurons in the PVN and OT fibers projecting to the NAc [166,169]. Activating CRF2s centrally decreases OT release in the NAc, whereas blocking CRF2s elevates OT release in this region [169]. Together, these findings suggest that pair bonding primes the CRF system and, upon partner separation, activity at CRF2s leads to diminished OT signaling in the NAc and subsequent passive stress-coping behaviors, which may reflect negative affect [2]. In a naturalistic, non-laboratory setting, when a pair-bonded vole in the wild leaves its partner to forage for food, this increase in CRF release may diminish OT release in the NAc over time, creating a negative affect. Much like when a person with a substance abuse disorder is without drugs, this subsequently motivates the vole to reunite with its partner, thus maintaining the bond [9].

Partner separation in humans may be short-term (e.g., when a partner is away for a trip), long-term (e.g., when partners live in different regions), or permanent (e.g., a breakup or death of a partner). Being away from a partner leads to rumination about that partner and a desire to be reunited, and longer-term separations are followed by psychological and physiological changes similar to those seen in prairie voles [170,171]. The loss of one’s partner is linked to the onset of a variety of psychiatric conditions, including major depressive episodes, panic disorder, and post-traumatic stress disorder [75]. Within 30 days of bereavement, there is an elevated risk for heart attack and stroke, along with alterations to immune function that are linked to the severity of anxiety and depressive symptoms [172,173] (Figure 2f). Over time, grief abates as individuals adapt to life without their partner; however, some individuals experience prolonged pathological grief, termed complicated grief (CG), characterized by intense yearning and recurrent painful emotions [174]. When viewing images of a deceased loved one, the brain regions associated with the pain network (ACC, IC, PAG) are activated in those with non-CG and CG alike [174]. In those with CG, viewing these images also results in increased activity in the NAc, and this activation is correlated with the degree of yearning for the loved one. This may explain why those with CG have difficulty adjusting their grief response over time, as thoughts about the loved one continue to activate DA-rich reward regions in the prolonged absence of the partner.

### 3.5. The Healing Power of Love

Love lost can pose risks to one’s health and well-being; however, evidence from rodent and human work suggests protective health benefits for being a part of a pair-bonded partnership. For example, married individuals, compared to unmarried individuals, are less likely to die following cancer diagnosis and heart failure, likely due, in part, to more involved care provided by a partner [76,175]. Additionally, individuals over 65 who are married have an increased life expectancy of ~2 years compared to non-married individuals [77]. Maintaining positive social relationships with others predicts a lower risk for physiological dysregulation, whereas social isolation increases the risk of inflammation and hypertension [1]. In monogamous Peromyscus mice, pair bonding has been shown to be protective against tumor growth [176]. Human lung cancer cells that were transplanted to pair-bonded mice showed less tumorigenesis compared to mice whose bond had been disrupted, contributing to evidence of the detrimental effects of bond disruption. Interestingly, tumors transplanted from pair-bonded mice to immunocompromised virgin mice displayed less growth than tumors originating from mice with disrupted bonds. This indicates that pair bond status affects cancer cell proliferation even in the absence of continued pair bond stimuli. Furthermore, transcriptomic analysis of these tumor cells revealed differential expression of genes linked to cell migration and tissue morphogenesis between pair-bonded and bond-disrupted mice. Although the mechanism of how pair-bonding leads to biochemical and genetic alterations in human cancer cells is unknown, it has been proposed that hormonal factors released from the pituitary gland in response to social stimuli may bind with tumor cells to alter morphology and growth [176]. OT, DA, AVP, and CRF all directly regulate secretions from the pituitary gland and are heavily implicated in pair bonding. Furthermore, OT is associated with lower inflammation, reduced microglial activation, and increased wound healing [177,178,179]. In patients with ovarian cancer, high levels of OT, both in the plasma and locally at tumors, are associated with lower levels of an inflammatory cytokine implicated in ovarian tumors [180]. Patients with high OT levels at the location of the tumor had a 34% decreased risk of death from the disease compared to patients with low OT levels. Together these findings point to the potential healing properties of pair bonding and sustained social interactions, which may be mediated in part by continued OT signaling from partner cues.

## 4. Conclusions

The experience of being “in love” may be uniquely human; however, rodent studies elucidate the neurobiological mechanisms underlying stages of bonding that are conserved across species. Pair bonds established between individuals rely on a series of processes to form and maintain that are facilitated by OT, DA, and AVP signaling. Pair bonding likely evolved from bonds between mothers and their offspring, with early-life parental nurturing setting the stage for later relationship development. There are several progressions of pair bonding, beginning with initial attraction to another individual and a choice to pursue that individual. The formation of a pair bond is facilitated by mating, and once a bond has been established, several behaviors emerge to prevent the disruption of that bond. Separation from a partner can lead to physiological and psychological signs of stress; however, evidence suggests an association between long-term companionship and positive health outcomes. Research in rodent models of social behavior allows for the use of more invasive and controlled techniques than are available in research with human subjects. For example, rodent research utilizes site- and cell-type-specific manipulations or recordings (e.g., pharmacological infusion, chemogenetics, optogenetics, electrophysiology) in conjunction with behavioral assays and post-mortem tissue analyses. This allows experimenters to induce alterations to intracellular signaling or to neural circuits and examine the direct effect of these manipulations on animals’ behavior during controlled behavioral paradigms. Rodent research also allows for the social history of the animal to be controlled, such as their early life parental care or their exposure to opposite-sex conspecifics. Human examinations of neural activity rely primarily on fMRI, which examines broad patterns of activation and deactivation in response to experimentally-presented stimuli and can be paired with self-report, psychological, and physiological measures. Although human studies involve less experimental control, self-report measures provide context to the emotional state and “feelings” of individuals during experiments that cannot be assessed in rodent models. When examining human experiences of love and long-term relationships, it is also imperative to also consider the socio-cultural factors at play. Romantic love is present in almost all human societies, and despite differences in acceptance of polygamy and extra-marital sex across cultures, monogamy remains the dominant marriage type even in groups where polygamy is accepted [13,181,182]. Furthermore, although humans are not sexually exclusive, the rates of extra-pair paternity are relatively low compared to other socially monogamous but not sexually exclusive animals, such as certain monogamous birds [182]. Attachment style, relationship history, and familial or cultural expectations work in tandem with neurobiological mechanisms to influence mate choice and romantic engagements [60,117,118,182]. Ultimately, rodent and human studies complement each other, with each providing unique insights into the processes underlying bonding. While this field of research is primarily concerned with understanding these processes from a basic knowledge perspective, studies provide useful conceptual frameworks to understand romantic relationships that can be implemented into interventions aimed at improving or prolonging relationships between individuals. Throughout history, humans have been mystified and inspired by love, and while there is still much to discover, technical advances in rodent and human research further our understanding of “what chemical forces flow from lover to lover”.

## Figures and Tables

**Figure 1 biology-12-00844-f001:**
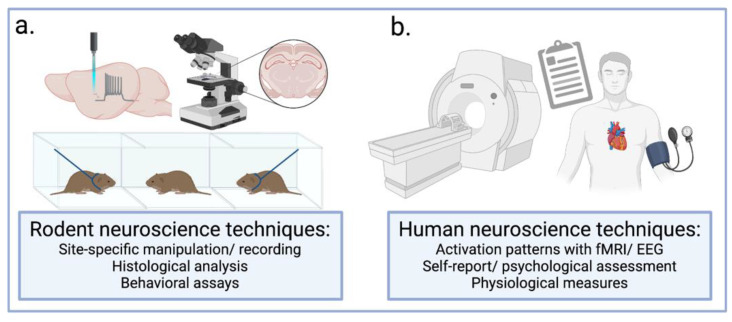
Techniques available for dissecting the neural underpinnings of behavior and emotion. (**a**) Rodent research is enhanced by the utilization of invasive techniques and behavioral assays that are not available for human use. Site-specific manipulations or recordings, such as chemogenetics, optogenetics, electrophysiology, pharmacological infusion, or viral knockdown, aid in determining the function of specific cellular populations. Post-mortem histological analysis allows for the visualization, segmentation, and quantification of neural substrates, including receptors and fibers, along with activity-induced protein expression. A wealth of assays have been developed to test for a range of behaviors and processes under experimentally controlled conditions, for example, the partner preference test (PPT) used to test the strength of pair bonds in prairie voles. Together, these techniques used in rodents allow for greater understanding of the neural mechanisms underlying processes such as pair bonding that can be translated to human work. (**b**) Research in humans involves less invasive techniques to dissect the neural networks underlying bonding. Neuroimaging through functional magnetic resonance imaging (fMRI) and electroencephalogram (EEG) allows for visualization of neural activity in response to experimentally-presented stimuli. fMRI and EEG can be paired with self-report, physiological, and psychological measures from subjects, providing further context for changes in neural activity captured through neuroimaging. Figure created with BioRender.com (accessed 4 June 2023).

**Figure 2 biology-12-00844-f002:**
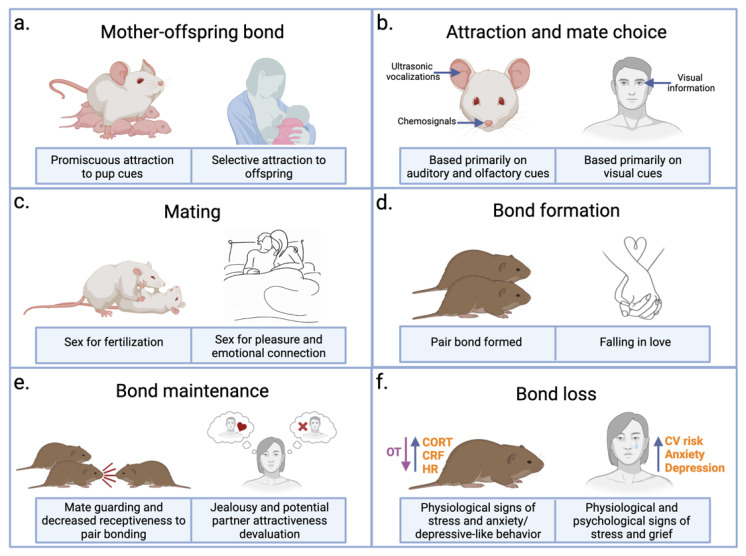
Comparative processes of bonding. (**a**) Rodent and human bonding are driven by shared processes across species. Mother–infant bonds are the evolutionary antecedent for pair bonding, relying on selective recognition of infant stimuli and persistent motivational drive to care for offspring. (**b**) Recognition of and attraction to potential mates in rodents uses primarily auditory and olfactory signals, whereas, in humans, potential partners are initially evaluated primarily based on visual information. (**c**) For rodents and humans alike, mating facilitates pair bond formation; however, human sexual behavior is complex, with individuals engaging in sexual activity for a variety of reasons, including for pleasure and to strengthen connections. (**d**) Once a bond has formed, monogamous prairie voles will prefer to mate and spend time with their partner, as well as engage in biparental care. Pair bonding in humans, referred to as “being in love” is accompanied by similar affiliative behaviors. (**e**) Long-term bonds take work to maintain, especially in the face of other potential mates. Prairie voles engage in mate guarding and have decreased receptiveness to extra-pair bond formations, allowing for bonds to remain stable over time. Humans experience feelings of jealousy and devalue the attractiveness of other individuals, both contributing to long-term bond maintenance. (**f**) The loss of a partner results in physiological and psychological risk factors across species. Rodents experience a reduction in oxytocin (OT) release accompanied by an increase in corticotropin-releasing factor (CRF), as well as elevated heart rate (HR). Loss of a loved one in humans results in elevated cardiovascular (CV) risk, increased experiences of anxiety or depression, and overall feelings of grief. Figure created with BioRender.com (accessed 8 June 2023).

**Figure 3 biology-12-00844-f003:**
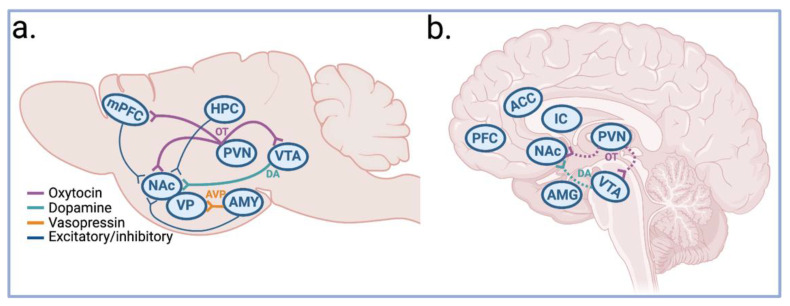
Comparison of brain regions involved in pair bonding and love across species. Similar brain regions are involved in pair bonding and romantic love across species. In rodents, the involvement of neural circuits and neuromodulators can be studied, whereas, in human fMRI, nodes of activation and deactivation are examined. (**a**) In prairie voles, oxytocin (OT) produced in the paraventricular nucleus of the hypothalamus (PVN) is released to medial prefrontal cortex (mPFC) and nucleus accumbens (NAc). Neural entrainment of the mPFC and NAc predicts pair bond formation. Representations of the partner are likely encoded by cells in the hippocampus (HPC) and the extended amygdala (AMY), regions that project to the NAc and may contribute to the rewarding properties of partner-related stimuli. The ventral tegmental area (VTA) releases dopamine (DA) to the NAc. Vasopressin (AVP) is released from the AMY to the ventral pallidum (VP) to facilitate pair bonding. (**b**) In humans, the NAc and VTA are activated upon viewing images of one’s partner, both regions in the mesolimbic DA system, and this activation is enhanced by IN-OT. Dotted lines represented inferred circuits based on neuroimaging studies in conjunction with IN-OT administration. Other regions activated include the insular cortex (IC), anterior cingulate cortex (ACC), and substantia nigra (SN). Deactivation is seen in the AMY and PFC. Figure created with BioRender.com (accessed 4 June 2023).

## Data Availability

Not applicable.

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
