# Peer review of "The Neurobiology of Love and Pair Bonding from Human and Animal Perspectives"

_biology, 2023, doi:10.3390/biology12060844_

Round 1

Reviewer 1 Report

This scientific publication entitled 'The neurobiology of love and pair bonding from human and animal perspectives' by Sarah A Blumenthal, Larry J Young investigated the neurobiological mechanisms involved in love and attachment in humans and animals. It provides an overview of the functions of oxytocin, dopamine and vasopressin in the management of the neural networks that underlie social bonding in humans and animals.

Although animal models can provide useful data on the neurobiological mechanisms involved in human behaviour, it is essential to consider the specific socio-cultural aspects that influence the connections and interactions between individuals.

Overall, although the analysis presents a beneficial perspective on the neurobiological processes that underpin love and emotional relationships, it is crucial to consider these remarks and to approach the subject with a multidisciplinary approach that takes into account biological and socio-cultural elements

Finally, it would be interesting to explore the practical implications of this research, for example how this knowledge could be used to improve human relationships or to treat attachment-related psychological disorders?

Author Response

We appreciate the reviewer’s opinion that the manuscript would benefit from the addition of information on socio-cultural factors influencing connections between individuals and have addressed these comments below:

  1. Although animal models can provide useful data on the neurobiological mechanisms involved in human behaviour, it is essential to consider the specific socio-cultural aspects that influence the connections and interactions between individuals. Overall, although the analysis presents a beneficial perspective on the neurobiological processes that underpin love and emotional relationships, it is crucial to consider these remarks and to approach the subject with a multidisciplinary approach that takes into account biological and socio-cultural elements.

We added the following sentences to lines 772-781 and have chosen not to go more into depth on socio-cultural factors as this would be out of the scope of the present paper:
“When examining human experiences of love and long-term relationships, it is also imperative to also consider the socio-cultural factors at play. Romantic love is present in almost all human societies and despite differences in acceptance of polygamy and extra-marital sex across cultures, monogamy remains the dominant marriage-type even in groups where polygamy is accepted [13,172,173]. Furthermore, although humans are not sexually exclusive, the rates of extra-pair paternity are relatively low compared to other socially monogamous but not sexually exclusive animals, such as certain monogamous birds [173]. Attachment style, relationship history, and familial or cultural expectations work in tandem with neurobiological mechanisms to influence mate choice and romantic engagements [55,108,109,173].”

  1. Finally, it would be interesting to explore the practical implications of this research, for example how this knowledge could be used to improve human relationships or to treat attachment-related psychological disorders?

We believe that the purpose of the animal studies on bonding are less about treatments and more so aimed at basic understanding of these processes; however, we added the sentences below:

The following sentence was added to lines 44-46: “These findings allow for greater understanding into our own human experiences of bonding and can inform strategies to engage in and maintain positive relationships.”

The following sentence added to lines 783-786: “While this field of research is primarily concerned with understanding these processes from a basic knowledge perspective, studies provide useful conceptual frameworks to understand romantic relationships that can be implemented into interventions aimed at improving or prolonging relationships between individuals.”

Reviewer 2 Report

I've read the paper titled "The neurobiology of love and pair bonding from human and animal perspectives"

I think the paper is well written and adds some precious insight to understand the biology of love and mating processes in both humans and animals. 

I have some suggestions for authors to improve their manuscript: 

Line 36: I know that the argument treated in the paper is quite common and we all know how love has been expressed in arts. However, I think it might be of interest to add some references to allow the reader to deepen the literature that has inspired the authors.

Line 44: citation needed

Line 46: citation needed

Line 54: citation needed

Line 93: citation needed

Line 96: citation needed

Line 107: Citation needed

Paragraph 2.1: It would be of interest to report biological data related to infanticide in both humans and animals.

Line 250: citation needed after “respectively”

Lines 265 to 274: citations needed.  

Lines 274 ti 276 citation needed

Lines 276 to 278: authors stated that “Research on the underlying neurobiological mechanisms of 276 mating behavior and pair bonding in prairie voles, as well as in other rodents, provides translational insights into human experiences of love”. I suggest being more cautious, “might provides translational…”

Line 326: citation needed to support that primary cue for humans is visual cue.

Line 331: “Mate choice is driven by initial attraction of potential 330 mates and evaluation of their reproductive fitness through various modalities[61,62].” It is not clear whether the authors are referring to humans or other animal species. At least in human the evaluation of reproductive fitness is a subjective perception. Furthermore, it should be noted that at least in humans attachment style play a role right from the beginning of a relationship, and also in mating choices. The attractiveness of another human beings could be only marginally driven by a real reproductive fitness in terms of physical condition or social status.

Paragraph 3.1: when discussing faces attractiveness it could be of interest to define if attractiveness is due to specific features that could be generally precepted as attractive or if the perception of an attractive face is only subjective and differs in each person. Also it could be of interest to define which face features pays a role in mating process and the mechanism related to face perception (very briefly).

Author Response

We are thankful for the reviewer’s comments that the paper adds “precious insight to understand the biology of love and mating processes in both humans and animals” and for their detailed comments. Please see the changes below based on these reviewer suggestions:

  1. I know that the argument treated in the paper is quite common and we all know how love has been expressed in arts. However, I think it might be of interest to add some references to allow the reader to deepen the literature that has inspired the authors.

Rather than citing articles, we provided some examples of pieces of art, literature, and architecture that have been inspired by love. The reader can explore on their own the stories behind these iconic pieces and how love is represented in them.

  1. Line 44: citation needed

We have added the following citations to line 44:

  • Bales, K.L.; Ardekani, C.S.; Baxter, A.; Karaskiewicz, C.L.; Kuske, J.X.; Lau, A.R.; Savidge, L.E.; Sayler, K.R.; Witczak, L.R. What is a pair bond? Horm Behav 2021, 136, 105062, doi:10.1016/j.yhbeh.2021.105062.
  • Rilling, J.K.; Young, L.J. The biology of mammalian parenting and its effect on offspring social development. Science 2014, 345, 771-776, doi:10.1126/science.1252723.

  1. Line 54: citation needed

We have added the following citations to line 54 (now line 56):

  • Bales, K.L.; Ardekani, C.S.; Baxter, A.; Karaskiewicz, C.L.; Kuske, J.X.; Lau, A.R.; Savidge, L.E.; Sayler, K.R.; Witczak, L.R. What is a pair bond? Horm Behav 2021, 136, 105062, doi:10.1016/j.yhbeh.2021.105062.
  • Lambert, C.T.; Sabol, A.C.; Solomon, N.G. Genetic Monogamy in Socially Monogamous Mammals Is Primarily Predicted by Multiple Life History Factors: A Meta-Analysis. Frontiers in Ecology and Evolution 2018, 6, doi:10.3389/fevo.2018.00139.

  1. Line 93: citation needed

We have added the following citations to line 93 (now line 95):

  • Walum, H.; Young, L.J. The neural mechanisms and circuitry of the pair bond. Nat Rev Neurosci 2018, 19, 643-654, doi:10.1038/s41583-018-0072-6.
  • Zablocki-Thomas, P.B.; Rogers, F.D.; Bales, K.L. Neuroimaging of human and non-human animal emotion and affect in the context of social relationships. Front Behav Neurosci 2022, 16, 994504, doi:10.3389/fnbeh.2022.994504.

  1. Line 96: citation needed

This sentence serves to set up what the review will be discussing (now line 99). We added citations to a few studies illustrating this point that will be discussed further later. We have also modified “dopamine (DA) to “the mesolimbic dopamine (DA) reward system” in this sentence.

  • Keebaugh, A.C.; Barrett, C.E.; Laprairie, J.L.; Jenkins, J.J.; Young, L.J. RNAi knockdown of oxytocin receptor in the nucleus accumbens inhibits social attachment and parental care in monogamous female prairie voles. Soc Neurosci 2015, 10, 561-570, doi:10.1080/17470919.2015.1040893.
  • Hung, L.W.; Neuner, S.; Polepalli, J.S.; Beier, K.T.; Wright, M.; Walsh, J.J.; Lewis, E.M.; Luo, L.; Deisseroth, K.; Dölen, G.; et al. Gating of social reward by oxytocin in the ventral tegmental area. Science 2017, 357, 1406-1411, doi:10.1126/science.aan4994.
  • Barrett, C.E.; Keebaugh, A.C.; Ahern, T.H.; Bass, C.E.; Terwilliger, E.F.; Young, L.J. Variation in vasopressin receptor (Avpr1a) expression creates diversity in behaviors related to monogamy in prairie voles. Horm Behav 2013, 63, 518-526, doi:10.1016/j.yhbeh.2013.01.005.
  • Scheele, D.; Wille, A.; Kendrick, K.M.; Stoffel-Wagner, B.; Becker, B.; Güntürkün, O.; Maier, W.; Hurlemann, R. Oxytocin enhances brain reward system responses in men viewing the face of their female partner. Proc Natl Acad Sci U S A 2013, 110, 20308-20313, doi:10.1073/pnas.1314190110.
  • Walum, H.; Westberg, L.; Henningsson, S.; Neiderhiser, J.M.; Reiss, D.; Igl, W.; Ganiban, J.M.; Spotts, E.L.; Pedersen, N.L.; Eriksson, E.; et al. Genetic variation in the vasopressin receptor 1a gene (AVPR1A) associates with pair-bonding behavior in humans. Proc Natl Acad Sci U S A 2008, 105, 14153-14156, doi:10.1073/pnas.0803081105.

  1. Line 107: Citation needed

We do not believe a citation is necessary here, as we are stating what we believe to be common knowledge. We are not aware of any specific scientific studies testing this, but also do not state that every mother experiences these feelings after birth.

  1. Paragraph 2.1: It would be of interest to report biological data related to infanticide in both humans and animals.

We believe commenting on infanticide may take away from the main message of the paper which focuses on the role of oxytocin, vasopressin, and dopamine in bonding. In many mammals, infanticide by males is a strategy believed to be employed in situations where a dependent offspring prevents the female from entering into estrus or becoming pregnant again (Lukas and Huchard, 2019). Infanticide thereby creates extra-reproductive opportunities for the male. In female mammals, infanticide is related to social structure and is more prevalent in cases where females breed in groups. Females engage in infanticide when access to resources for her offspring is threatened by competition from other females’ offspring. Alternatively, human infanticide is often caused by post-partum depression or psychosis in mothers or results from lack of access to abortion for unwanted pregnancy (Moseson et al., 2019; Brockington, 2017). There are also no known studies on the neural mechanisms underlying infanticide in humans. Although this is an interesting avenue to consider, we believe this is beyond the scope of the paper and would take away from the flow of the section on parental bonding.

Citations for above paragraph:

  • Lukas D, Huchard E. The evolution of infanticide by females in mammals. Philos Trans R Soc Lond B Biol Sci. 2019 Sep 2;374(1780):20180075. doi: 10.1098/rstb.2018.0075. Epub 2019 Jul 15. PMID: 31303157; PMCID: PMC6664130.
  • Moseson H, Ouedraogo R, Diallo S, Sakho A. Infanticide in Senegal: results from an exploratory mixed-methods study. Sex Reprod Health Matters. 2019 Dec;27(1):1624116. doi: 10.1080/26410397.2019.1624116. Erratum in: Sex Reprod Health Matters. 2019 May;27(2):1637164. PMID: 31533577; PMCID: PMC7888053.
  • Brockington I. Suicide and filicide in postpartum psychosis. Arch Womens Ment Health. 2017 Feb;20(1):63-69. doi: 10.1007/s00737-016-0675-8. Epub 2016 Oct 24. PMID: 27778148; PMCID: PMC5237439.

  1. Line 250: citation needed after “respectively”

The following citation was added to line 250 (now line 256).

  • Ahern, T.H.; Olsen, S.; Tudino, R.; Beery, A.K. Natural variation in the oxytocin receptor gene and rearing interact to influence reproductive and nonreproductive social behavior and receptor binding. Psychoneuroendocrinology 2021, 128, 105209, doi:10.1016/j.psyneuen.2021.105209.

  1. Lines 265 to 274: citations needed.

This paragraph serves to set up what the following section will be discussing. We added citations to a few studies illustrating this point that will be discussed further later to what is now lines 272-280.

  • Rosenthal, G.G. Mate choice: the evolution of sexual decision making from microbes to humans; Princeton University Press: 2017.
  • Cooper, J.C.; Dunne, S.; Furey, T.; O'Doherty, J.P. Dorsomedial prefrontal cortex mediates rapid evaluations predicting the outcome of romantic interactions. J Neurosci 2012, 32, 15647-15656, doi:10.1523/jneurosci.2558-12.2012.
  • Fisher, H.; Aron, A.; Brown, L.L. Romantic love: an fMRI study of a neural mechanism for mate choice. J Comp Neurol 2005, 493, 58-62, doi:10.1002/cne.20772.
  • Keyes, K.M.; Pratt, C.; Galea, S.; McLaughlin, K.A.; Koenen, K.C.; Shear, M.K. The burden of loss: unexpected death of a loved one and psychiatric disorders across the life course in a national study. Am J Psychiatry 2014, 171, 864-871, doi:10.1176/appi.ajp.2014.13081132.

  1. Lines 274 to 276 citation needed

The following citations were added to 274-276 (now 280-282):

  • Chen, Z.H.; Yang, K.B.; Zhang, Y.Z.; Wu, C.F.; Wen, D.W.; Lv, J.W.; Zhu, G.L.; Du, X.J.; Chen, L.; Zhou, G.Q.; et al. Assessment of Modifiable Factors for the Association of Marital Status With Cancer-Specific Survival. JAMA Netw Open 2021, 4, e2111813, doi:10.1001/jamanetworkopen.2021.11813.
  • Jia, H.; Lubetkin, E.I. Life expectancy and active life expectancy by marital status among older U.S. adults: Results from the U.S. Medicare Health Outcome Survey (HOS). SSM Popul Health 2020, 12, 100642, doi:10.1016/j.ssmph.2020.100642.

  1. Lines 276 to 278: authors stated that “Research on the underlying neurobiological mechanisms of 276 mating behavior and pair bonding in prairie voles, as well as in other rodents, provides translational insights into human experiences of love”. I suggest being more cautious, “might provides translational…”

Sentence (now on lines 282-284) has been changed to “Research on the underlying neurobiological mechanisms of mating behavior and pair bonding in prairie voles, as well as in other rodents, may provide useful translational insights into human experiences of love.”

  1. Line 326: citation needed to support that primary cue for humans is visual cue.

The citation below has been added to line 326 (now line 307).

  • Roth, T.S.; Samara, I.; Kret, M.E. Multimodal mate choice: Exploring the effects of sight, sound, and scent on partner choice in a speed-date paradigm. Evolution and Human Behavior 2021, 42, 461-468.

  1. Line 331: “Mate choice is driven by initial attraction of potential 330 mates and evaluation of their reproductive fitness through various modalities[61,62].” It is not clear whether the authors are referring to humans or other animal species. At least in human the evaluation of reproductive fitness is a subjective perception. Furthermore, it should be noted that at least in humans attachment style play a role right from the beginning of a relationship, and also in mating choices. The attractiveness of another human beings could be only marginally driven by a real reproductive fitness in terms of physical condition or social status.

The sentence in line 331 (now line 311) was changed to “In animals, mate choice is driven by initial attraction of potential mates and evaluation of their reproductive fitness through various modalities [67,77].” We also added the following sentence to line 407-410: “Importantly, attraction and mate choice in humans is subjective and often influenced by socio-cultural expectations for relationships, early life experiences, and previous relationship experiences [55,108,109].”

  1. Paragraph 3.1: when discussing faces attractiveness it could be of interest to define if attractiveness is due to specific features that could be generally precepted as attractive or if the perception of an attractive face is only subjective and differs in each person. Also it could be of interest to define which face features pays a role in mating process and the mechanism related to face perception (very briefly).

The following sentences were added to section 3.1 (line 373-379) “Although perceived attractiveness is subjective and varies by culture, certain features are generally favored and may point to evolutionarily driven preferences to features indicative of reproductive fitness. For example, features favored include youthfulness, symmetry, and averageness (how closely a face resembles the majority of other faces) which are hypothesized to be related to the genetic diversity and health of the individual[93]. Facial attractiveness has also been found to be predictive of longevity and reproductive success[94,95].”

Reviewer 3 Report

The review by Blumenthal and Young provides a logical and straightforward framework for the phases of pair bond formation, from mate attraction to long-term bonding. By identifying parallels between rodent and human behavior and neurobiology, the authors facilitate the translation of results from animal models to human applications. The didactic breakdown of the formation of long-term social bonds into separate steps is particularly useful for researchers studying social behavior. The authors successfully deconstruct the complex phenomenon of love into simpler, manageable, and researchable components, which have been studied in rodents. The thorough and exhaustive review of the human and rodent literature on the topic serves not only as a helpful introduction for young researchers but also as a great summary of the current state of research on pair bonds for those working in this field.

The ambitious title, together with the evolutionary approach mentioned in the abstract (lines 14-15, 19-20), suggests a wider scope, possibly including other vertebrate taxa, in the review compared to the authors' stated aim in lines 93-95. I suggest rephrasing the sentence " We discuss the neural circuits and neuromodulators driving bonding across species, with animal studies providing insight into our human experiences of love.„ " to restrict the scope to mammals, possibly focusing on rodents.

Since most of the brain regions mentioned in the article are involved in the so-called social brain network or the mesolimbic reward circuit, it might be worth mentioning the conceptual background of their roles in regulating social behavior (e.g. https://pubmed.ncbi.nlm.nih.gov/15885690/ and https://pubmed.ncbi.nlm.nih.gov/10415653/). To provide context for the social salience network, a brief explanation of how it is related to the previously described social brain network or social decision-making network would be helpful. What are the differences and overlaps between these conceptual frameworks?

Additionally, mentioning the tactile modality would complement the detailed discussion on visual, olfactory, and auditory social cues around lines 350-360 (e.g., https://pubmed.ncbi.nlm.nih.gov/30009845/ ,  https://pubmed.ncbi.nlm.nih.gov/36524176/).

Minor issues:

Line 167: IN-OT appears first but the abbreviation is resolved only later (line 171)

Line 378: The resolution of the abbreviation ’AI’ could not be found.

There is the abbreviation AMG instead of AMY in Figure 3b.

Lines 506-509. The sentence ’Furthermore, investigations into OXTR and AVPR1a expression across monogamous and non-monogamous species provides cross-species comparisons for mechanisms driving pair bonding.’ could use some references. (possibly https://academic.oup.com/mbe/article/27/6/1269/1110683 and/or https://www.ncbi.nlm.nih.gov/pmc/articles/PMC7881006/ )

Line 593: Resolution for NHP (non-human primate?) is missing.

Author Response

We appreciate the reviewer’s comments that the review “serves not only as a helpful introduction for young researchers but also as a great summary of the current state of research on pair bonds for those working in this field” as this was our hope when constructing the manuscript. Their suggestions, both major and minor, improve the quality of the work and have been incorporated as detailed below.

  1. The ambitious title, together with the evolutionary approach mentioned in the abstract (lines 14-15, 19-20), suggests a wider scope, possibly including other vertebrate taxa, in the review compared to the authors' stated aim in lines 93-95. I suggest rephrasing the sentence " We discuss the neural circuits and neuromodulators driving bonding across species, with animal studies providing insight into our human experiences of love.„ " to restrict the scope to mammals, possibly focusing on rodents.

We have changed the sentence in lines 14-15 to now read “We discuss the neural circuits and neuromodulators driving bonding, with rodent models providing insight into our human experiences of love”.

  1. Since most of the brain regions mentioned in the article are involved in the so-called social brain network or the mesolimbic reward circuit, it might be worth mentioning the conceptual background of their roles in regulating social behavior (e.g. https://pubmed.ncbi.nlm.nih.gov/15885690/and https://pubmed.ncbi.nlm.nih.gov/10415653/). To provide context for the social salience network, a brief explanation of how it is related to the previously described social brain network or social decision-making network would be helpful. What are the differences and overlaps between these conceptual frameworks?

We have added the following sentences to lines 142-147 to introduce the mesolimbic DA system: “Following birth, activity in the mesolimbic dopamine (DA) system drives continuous interactions with offspring, including retrieval, licking, and grooming of pups [29,30]. This system comprises DA-producing neurons in the substantia nigra (SN) and the ventral tegmental area (VTA) that project to the striatum, cortex, limbic regions, and the hypothalamus to influence motivation, emotion, and reward-related behaviors[31].”

The following sentences to were added to the opening paragraph of section 3 (lines 285-297) to introduce the social brain network and the social decision making network: “Importantly, a considerable amount of overlap exists in the brain regions and neuromodulators involved in these processes in rodents and humans. Many of these regions are a part of established networks known to be involved in social behavior across species, such as the social brain network (SBN) and the social decision-making network (SDMN). The SBN comprises hypothalamic structures as well as the periaqueductal gray (PAG), medial amygdala (MeA), bed nucleus of the stria terminalis (BNST), and the LS that are reciprocally connected with the other regions of the network, contain receptors for gonadal hormones, and are implicated in at least one social behavior [73,74]. This network was later expanded as the SDMN to include structures in the mesolimbic DA system involved in detecting and appraising the rewarding aspects of social situations [75]. The involvement of these conversed networks of social behavior across species in attraction, mating, and bonding further illustrates the translational applicability of rodent research investigating these processes, which will be discussed further in this section”

  1. Additionally, mentioning the tactile modality would complement the detailed discussion on visual, olfactory, and auditory social cues around lines 350-360 (e.g., https://pubmed.ncbi.nlm.nih.gov/30009845/,  https://pubmed.ncbi.nlm.nih.gov/36524176/).

The following sentences were added to lines 354-361 to address somatosensation in mate choice in rodents: “In addition to olfactory and auditory cues, somatosensory cues detected through social tactile stimulation play an important role in interactions between animals. Social touch, as opposed to non-social touch, results in stronger responses from cells within the rodent barrel cortex, a division of the primary somatosensory cortex corresponding to processing signals emerging from whiskers[86]. Social touch also leads to the release of OT from the PVN and DA from the VTA to the NAc, resulting in conditioned place preference for social touch contexts and suggesting reinforcing properties of social tactile stimulation[87,88].”

The following sentences were added to the end of 3.1 (lines 397-404). “Although visual information is utilized over other sensory modalities for initial attraction, auditory and tactile cues carry important information about potential mates. Humans can typically decode the gender of an individual from their voice and distinguish between voices [103,104]. Social touch, on the other hand, conveys information about intimacy, warmth, and sexuality, with the frequency and bodily location of social touch indicating the closeness between individuals [105]. Both auditory and tactile social cues rely on the involvement of the mPFC and insula, implicated in socio-emotional and socio-salience processing [106,107].”

  1. Line 167: IN-OT appears first but the abbreviation is resolved only later (line 171)

IN-OT was defined placed after its first occurrence in line 173.

  1. Line 378: The resolution of the abbreviation ’AI’ could not be found.

AI (anterior insular cortex) was defined in line 385

  1. There is the abbreviation AMG instead of AMY in Figure 3b.

The abbreviation in the figure now reads “AMY”.

  1. Lines 506-509. The sentence ’Furthermore, investigations into OXTR and AVPR1a expression across monogamous and non-monogamous species provides cross-species comparisons for mechanisms driving pair bonding.’ could use some references. (possibly https://academic.oup.com/mbe/article/27/6/1269/1110683and/or https://www.ncbi.nlm.nih.gov/pmc/articles/PMC7881006/ )

The following citations were added to line 509 (now 533):

  • Insel, T.R.; Shapiro, L.E. Oxytocin receptor distribution reflects social organization in monogamous and polygamous voles. Proc Natl Acad Sci U S A 1992, 89, 5981-5985, doi:10.1073/pnas.89.13.5981.
  • Neural correlates of mating system diversity: oxytocin and vasopressin receptor distributions in monogamous and non-monogamous Eulemur. Sci Rep 2021, 11, 3746, doi:10.1038/s41598-021-83342-6.
  • Turner, L.M.; Young, A.R.; Römpler, H.; Schöneberg, T.; Phelps, S.M.; Hoekstra, H.E. Monogamy evolves through multiple mechanisms: evidence from V1aR in deer mice. Mol Biol Evol 2010, 27, 1269-1278, doi:10.1093/molbev/msq013.

  1. Line 593: Resolution for NHP (non-human primate?) is missing.

The abbreviation for NHP (non-human primate) is defined on line 365.

Round 2

Author Response

thanks